# An Efficient, Simultaneous Electrochemical Assay of Rosuvastatin and Ezetimibe from Human Urine and Serum Samples

**DOI:** 10.3390/mps5060090

**Published:** 2022-11-01

**Authors:** Leyla Karadurmus, Sevinc Kurbanoglu, Bengi Uslu, Sibel A. Ozkan

**Affiliations:** 1Department of Analytical Chemistry, Faculty of Pharmacy, Adıyaman University, Adıyaman 02040, Turkey; 2Department of Analytical Chemistry, Faculty of Pharmacy, Ankara University, Ankara 06560, Turkey

**Keywords:** rosuvastatin, ezetimibe, glassy carbon electrode, adsorptive stripping differential pulse voltammetry

## Abstract

The drug combination of rosuvastatin (ROS) and ezetimibe (EZE) is used to treat hypercholesterolemia. In this work, a simultaneous electrochemical examination of ROS and EZE was conducted for the first time. The electrochemical determination of ROS and EZE was carried out using adsorptive stripping differential pulse voltammetry (AdSDPV) on a glassy carbon electrode (GCE) in 0.1 M H_2_SO_4_. The effects of the pH, scan rate, deposition potential, and time on the detection of ROS and EZE were analyzed. Under optimum conditions, the developed sensor exhibited a linear response between 1.0 × 10^−6^ M and 2.5 × 10^−5^ M for EZE and 5.0 × 10^−6^ M, and 1.25 × 10^−5^ M for ROS. The detection limits for ROS and EZE were 3.0 × 10^−7^ M and 2.0 × 10^−6^ M, respectively. The developed sensor was validated in terms of linear range, accuracy, precision, the limit of determination (LOD), and the limit of quantification (LOQ), and it was evaluated according to ICH Guidelines and USP criteria. The proposed method was also used to determine ROS and EZE in human urine and serum samples, which are reported in terms of recovery studies.

## 1. Introduction

Rosuvastatin (ROS) is a hypercholesterolemia drug that lowers plasma cholesterol levels (Figure 1a) [1]. ROS has a structure that is similar to most other synthetic statins, but unlike other statins, it contains sulfur. ROS is a competitive inhibitor of the enzyme HMG-CoA reductase [2,3,4]. Ezetimibe (EZE) is a drug that the FDA has confirmed as curing hypercholesterolemia (Figure 1b). EZE is the first lipid-lowering drug that reduces the amount of lipoprotein cholesterol by preventing the absorption of cholesterol at the brush-border level of the intestine. It prevents the intestinal uptake of dietary and bile cholesterol [4,5].

Statins and EZE have different lipid-lowering mechanisms of action, and combining them can obtain the strongest impact on lowering lipids and stabilizing plaque areas [6]. In the literature, it has been found that the combination of ROS and EZE further lowers total cholesterol and LDL cholesterol, clearly lowering triglyceride levels, and potentiating the lipid-lowering effects. The combination of ROS and EZE decreases lipid levels and the plaque burden. The combination of a statin and EZE has a greater effect on coronary plaque regression in patients with acute coronary syndrome [6,7]. Adding EZE to ROS significantly improves many more lipid parameters than does doubling the ROS dose [8]. The literature includes descriptions of patients who received 5, 10, 20, or 40 mg of ROS every day, and the average plasma concentration for ROS was 1.6 ng/mL, 3.5 ng/mL, 6.3 ng/mL, and 9.8 ng/mL, respectively [9]. For patients taking one dose of 10 mg of ezetimibe, average ezetimibe peak plasma concentrations (C_max_) of 3.4 to 5.5 ng/mL were acquired within 4 to 12 h [10].

The two major fields of the natural sciences, chemistry and electrical science, came together in the 19th century to form electrochemistry [11]. Electrochemical techniques are extensively used in drug analysis. Among all of the electrochemical methods, stripping analysis is one of the most sensitive electrochemical techniques, and it is therefore used in quantitative determinations, especially in drug analysis. In recent years, stripping voltammetry has been used in the analysis of many drug substances [12]. The reason for this great sensitivity is the combination of an efficient accumulation phase with advanced measurement processes that produce an excellent signal [13,14]. The adsorptive accumulation is intended to deposit the analyte present in the solution on an electrode surface with a small surface area. Stripping voltammetry is also used in clinical practice and allows the conduct of various analyses of human blood, urine, and tissues [15].

The literature reveals some analytical techniques for the simultaneous detection of ROS and EZE. These methods are reverse-phase high-performance liquid chromatography [16,17], micellar liquid chromatography [18], high-performance column liquid chromatography, high-performance thin-layer chromatography [19], spectrophotometry [20,21], and liquid chromatography/mass spectrometry [22,23]. In this work, EZE and ROS were electrochemically analyzed using the AdSDPV technique at GCE. The efficacy of the electrochemical method was fully analyzed for the detection of ROS and EZE in commercial human serum and in urine samples, and we report on it in terms of recovery studies.

## 2. Experimental Design

### 2.1. Materials

Different supporting electrolytes of H_2_SO_4_ solutions (0.1 and 0.5 M), acetate (pH 3.7–5.7), and phosphate (pH 2.0–8.0) buffers were prepared for electrochemical measurements. AdSDPV voltammogram recordings were obtained after the addition of each aliquot. Drug-free human serum from male AB plasma was purchased from Sigma-Aldrich (St. Louis, MO, USA). Acetic acid, acetonitrile, methanol, phosphoric acid, sodium acetate trihydrate, sodium dihydrogen phosphate dihydrate, sodium hydroxide, sodium phosphate monobasic, sodium phosphate, and sulfuric acid were purchased from Sigma-Aldrich. All reagents were of analytical grade and were used without pre-processing. All measurements were realized at room temperature; all solutions were kept from light and used within 24 h to prevent degradation.

### 2.2. Equipment

A Bioanalytical Systems (BAS 100W) electrochemical analyzer with a standard three-electrode system was used for the voltammetric measurements. The three-electrode system included a platinum-wire counter electrode, an Ag/AgCl-saturated KCl reference electrode, and a GCE (GC, BAS; 3 mm, diameter), which served as a working electrode. The surface of the GCE was polished with an aqueous slurry of alumina powder (Φ: 0.01 μm) on a damp, smooth polishing cloth just before each experiment. The pH was checked using a pH meter Model 538 (Weilheim, Germany). Operating conditions for AdSDPV were as follow: pulse amplitude, 50 mV; deposition time, 15 s; scan rate, 20 mV/s; pulse width, 50 ms; sensitivity, 10 µA/V; sample width, 17 ms; pulse period, 200 ms; quiet time, 10 s.

## 3. Procedures

### 3.1. Standards and Sample Preparation

The 1 × 10^−3^ M stock solution of ROS and EZE was prepared in methanol and kept in a refrigerator (+4 °C). The solutions of ROS and EZE for the voltammetric measurements were prepared by direct dilution of the stock solution with 0.1 M H_2_SO_4_, and they included a constant amount of methanol (20%, *v*:*v*). Analytical curves were obtained by adding aliquots of the stock solutions of ROS and EZE into the electrochemical cell containing 10.0 mL of the 0.1 M H_2_SO_4_ with a constant amount of methanol.

### 3.2. Biological Sample Preparation

The applicability of the developed procedure to human urine samples was also investigated. Drug-free urine samples were collected from a healthy laboratory employee on the day of the experiment. To prepare a stock urine solution, 5.4 mL of acetonitrile, 3.6 mL of the drug-free urine samples, and 1 mL of the ROS/EZE stock solution (1 × 10^−3^ M) were placed in a 10 mL centrifuge tube. First, the mixture was vortexed for 10 min, and then it was centrifuged at 3500 rpm for 30 min. The supernatant part was carefully transferred to a distinct, clean tube. In this procedure, acetonitrile acted as a precipitating agent. A ROS/EZE-free sample of the same urine was used as a blank solution. All measurements were performed at least in triplicate, and the standard addition technique was performed for the determination of ROS/EZE.

Synthetic human serum was kept frozen at −20 °C in a freezer until analysis. For the preparation of a stock serum sample, a standard procedure was followed. Quantities of 1 mL of ROS/EZE, 5.4 mL of acetonitrile, and 3.6 mL of synthetic human serum were added to a centrifuge tube to prepare a stock serum solution. First, it was vortexed for 10 min and then centrifuged at 3500 rpm for 30 min, and later, the supernatant was taken. Here, acetonitrile was used to precipitate serum proteins. The supernatant was diluted with 0.1 M H_2_SO_4_ to prepare certain concentrations for the recovery measurements. All of the experiments were performed at least three times for calibration and five times for the recovery experiments.

Analytical curves were obtained by adding aliquots of the stock solutions of ROS and EZE from synthetic human serum or human urine into the electrochemical cell containing 10.0 mL of the 0.1 M H_2_SO_4_ with a constant amount of methanol.

## 4. Results and Discussion

### 4.1. Voltammetric Behavior of ROS and EZE

The voltammetric behavior of ROS and EZE was examined on a GCE in detail. In the first step, the behavior of ROS and EZE was investigated by CV studies to characterize their electrochemical oxidation behavior in the range of 0 V to 1.6 V. The CV results indicated the irreversible nature of the oxidation process of ROS and EZE. Moreover, the adsorptive stripping differential pulse voltammetric (AdSDPV) technique was further used, and the anodic oxidation was observed until reaching a potential of about 0.9 V, and 1.2 V; there was a single well-defined and sharp oxidation peak for EZE and ROS, respectively, using the AdSDPV technique on a GCE in 0.1 M H_2_SO_4_ (Figure 1).

### 4.2. Influence of the pH

The electrochemical behavior of ROS and EZE was studied within a wide pH range (pH 0.3–7.0) using the DPV technique on a GCE. With the DPV method, the maximum current occurred in the 0.1 M H_2_SO_4_ medium. The following equation followed the effect of pH on the peak potential. The *E_p_*-pH plots indicated that a pH increase caused the shifting of peak potentials to less positive values (Figure 2).
*E_p_* (mV) = 1354.24 − 22.79 pH; R^2^ = 0.997 for ROS
*E_p_* (mV) = 998.49 − 50.99 pH; R^2^ = 0.998 for EZE

### 4.3. Influence of the Scan Rate

Scan rate experiments were performed to understand the electrochemical oxidation/reduction mechanisms, such as adsorption or diffusion. The influence of the scan rate between 5 and 1000 mV/s on the peak current and potential was investigated in 0.1 M H_2_SO_4_ using CV, where the highest peak was obtained in pH studies using a GC electrode. 

The plot of *E_p_* vs. log *v* was linear; this attitude is coherent with the EC nature of the reaction in which the electrode reaction is coupled with an irreversible follow-up chemical step in CV. According to [24], *E_p_* can be defined by the following equation;
$$E_{p}=E^{0’}-\frac{2.303RT}{\alpha nF}\log\frac{RTk^{0}}{\alpha nF}+\frac{2.303RT}{\alpha nF}\log v$$
where *E*^0^ is the formal potential, *R* is the gas constant, *T* is the temperature, *k*^0^ is the standard heterogeneous rate constant, α is the transfer coefficient of the oxidation of ROS and EZE, *v* is scan rate, *F* is the Faraday constant, and n is the number of electrons that are involved in the electrooxidation of ROS and EZE [22].

In general, *α* is used as 0.5 for irreversible processes. Since *α* is 0.5 for irreversible systems, n can be calculated from

*E_p_* (V) = 0.046 log *v* (V·s^−1^) + 1.312 (r = 0.997) (0.1 × 10^−3^ M ROS), and n is found to be 2.36 for ROS, and

*E_p_* (V) = 0.049 log *v* (V·s^−1^) + 1.066 (r = 0.997) (0.1 × 10^−3^ M EZE), and n was calculated as being 2.38 for EZE (Figure 3a,b).

Moreover, the logarithm of peak current vs. the logarithm of scan rate gives more detailed information about the electrochemical mechanisms. When these graphs were plotted, for EZE, from the slope of the equation log (*I_p_*) = 0.783 log *v* − 1.186 (r = 0.998), it can be concluded that the reaction is adsorption-controlled since the slope was close to 1. Thus, as a result of the scan rate experiments, in the 0.1 M H_2_SO_4_ medium, the electrochemical behavior of EZE was found to be adsorption-controlled (Figure 3d).

For ROS, the slope of the equation log (*I_p_*) = 0.582 log *v* − 1.008 (r = 0.992), and the electrochemical behavior of ROS was found to be diffusion-controlled (Figure 3c). As we aimed to determine these two drug-active compounds simultaneously, we applied the adsorptive stripping method, which enabled us to assess ROS and EZE precisely.

In the literature, the electrochemical determination of ROS and its possible oxidation mechanism have been studied. The authors suggested an electrooxidation mechanism involving a Kolbe electrolysis reaction of the carboxylic acid group localized at the dihydroxyhept-6-enoic acid portion of the rosuvastatin calcium molecule [25,26,27]. In the literature, the electrochemical behavior and possible oxidation mechanism of EZE was also reported by the authors as being due to the inductive effect of the fluoride group in the aromatic rings of the EZE molecule; oxidation takes place in the hydroxyl group of phenol (EC mechanism) and the main voltammetric behavior of aromatic hydroxyl derivatives, which are structurally related to the mechanism of oxidation of EZE, may be postulated by the oxidation of the hydroxyl group on the aromatic ring [28,29].

### 4.4. Effect of Deposition Time and Potential

Parameters, such as deposition time and potential, significantly affect the AdSDPV peaks of the analytes. Hence, these parameters as related to AdSDPV were optimized to obtain the best results for the determination of ROS and EZE. The effect of the deposition time on stripping peak current was studied in the range of 0 s to 50 s, with 0 V deposition potential. It was observed that the peak current increased between 0 and 15 s (Figure 4). However, after 15 s, a decrease was observed in the peak current. As a result, 15 s was selected as the optimum deposition time. The effect of deposition potential, which is another important parameter, on stripping peak currents was studied in deposition potentials ranging from −0.1 V to +0.1 V, with a constant accumulation time of 15 s (Figure 4). A decrease in stripping peak currents was observed after the 0 V deposition potential, with an accumulation time of 15 s. A deposition time of 15 s and a deposition potential of 0 V, at which the maximum peak current was observed, were used in all subsequent experiments (Figure 4).

### 4.5. Analytical Characterization and Validation

Under optimum deposition potential and time conditions using the AdSDPV, samples with increasing concentrations of EZE and ROS were prepared. Analytical characterization in terms of LOD and LOQ based on 3 s/m and 10 s/m, respectively, were achieved using linear curves; where m is the slope of the related calibration curves and s is the standard deviation of the peak currents of the lowest concentration of the analyte. EZE was determined in the linear range between 1.0 × 10^−6^ M to 2.5 × 10^−5^ M, with a LOD of 3.0 × 10^−7^ M and a LOQ of 1.0 × 10^−6^ M. ROS was determined in the linear range between 5 × 10^−6^ M to 1.25 × 10^−5^ M, with a LOD of 2.0 × 10^−6^ M and a LOQ of 6.6 × 10^−6^ M. For the validation of the developed method, accuracy and precision were investigated by analyzing five replicate experiments between days and within days. Relative standard deviations (RSD%) were determined to control the precision of the technique. As summarized in Table 1, the results after statistical evaluation indicate that the technique is analytically acceptable (Figure 5 and Table 1).

### 4.6. Determination of Ezetimibe and Rosuvastatin in Biological Samples

In optimized conditions, the electrochemical method was also applied for the detection of EZE and ROS in buffer, spiked human serum, and urine samples, and reported in terms of recovery. Using the suggested method, the purified samples were used for the simultaneous determination of EZE and ROS. Recovery studies were performed by adding ROS and EZE in certain amounts to the human urine samples and serum samples by the proposed technique. The recovery studies of ROS and EZE were assessed based on the data given in Table 1. The proposed technique of RSD% and the average recovery results confirmed suitable accuracy and precision. The applicability of the developed method was indicated by constituting calibration graphs for ROS and EZE in the presence of spiked urine and serum samples. The developed technique was used for the accurate determination of ROS and EZE in biological samples without any pretreatment procedure. The outcomes of the calibration calculations and related parameters obtained in human urine and serum samples are given in Table 1. Recovery results of ROS and EZE were controlled with the corresponding calibration equations, obtained in human urine and serum samples, and found acceptable (Table 2). All results indicated the potential applicability of the developed method for evaluating human urine and serum samples.

## 5. Conclusions

In this study, the electrochemical behavior of ROS and EZE was studied simultaneously for the first time. AdSDPV was used for the reliable detection of ROS and EZE in a 0.1 M H_2_SO_4_ solution with commercial deproteinated human serum samples and human urine samples using a GCE, and results were reported in terms of recovery. The developed simple and low-cost method showed high sensitivity, a low limit of detection, good repeatability, and good linearity. In the proposed technique, we monitored linear relationships varying from 1.0 × 10^−6^ M to 2.5 × 10^−5^ M for EZE concentrations and from 5.0 × 10^−6^ M to 1.25 × 10^−5^ M for ROS concentrations. LOD values were found for ROS and EZE as 3.0 × 10^−7^ M and 2.0 × 10^−6^ M, respectively. As is stated in the literature, for patients taking 40 mg ROS daily, the average plasma concentration of ROS (C_max_) was 9.8 ng/mL (0.0098 µg/mL) [9]. Furthermore, after one dose of EZE, average EZE peak plasma concentrations (C_max_) of 3.4 to 5.5 ng/mL (0.0055 µg/mL) were obtained within 4 to 12 h (T_max_) [10]. These values are higher than our limit of detection value, indicating that the proposed method can be used to detect ROS and EZE in real samples. 

## Data Availability

Not applicable.

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
