# Peer review of "An Efficient, Simultaneous Electrochemical Assay of Rosuvastatin and Ezetimibe from Human Urine and Serum Samples"

_mps, 2022, doi:10.3390/mps5060090_

Round 1

Reviewer 1 Report

The paper can be published after minor revision reflecting comments inserted as yellow notes into attached manuscript

Author Response

We would like to thank the Reviewer for her/his comments and suggestions. Changes requested by the reviewer appear in yellow in the manuscript.

Reviewer 2 Report

The manuscript by Karadurmus et al., entitled “An Efficient Simultaneous Electrochemical Assay of Rosuvastatin and Ezetimibe from Human Urine and Serum Samples” is interesting and can be accepted for publication after major revision

[1]   This study report simultaneous determination of two drugs but I could not see the AdsDPV response of two drugs simultaneously. Authors must show this response since they claim its simultaneous determination.

[2]   The present work does not report the modification of GCE with suitable modifier such as nanomaterials. Authors must state suitable reason in the introduction.

[3]   Authors used 0.1 M H2SO4 as an electrolyte. What is the response of these drugs in some buffers like PBS, BR, etc. Since this study reports the analysis of biological samples, H2SO4 medium is not recommended.

[4]   What is the selectivity of this sensor in the presence of potential interferences such as glucose, ascorbic acid, uric acid, and mixture of interferences.

[5]   The scan rate study figure is still missing in the main text to confirm the adsorption control behaviour of the drugs at the electrode surface.

[6]   The sensitivity value is missing which must be reported.

[7]   The electrochemical reactions of these drugs at the surface of electrode are missing. Author must discus suitable reaction mechanism with chemical reactions.

Author Response

Reviewer 2

We would like to thank the Reviewer for her/his comments and suggestions. Changes requested by the reviewer appear in yellow in the manuscript.

Round 2

Reviewer 2 Report

Authors have addressed all the concerns raised, therefore, manuscript can be accepted for publication